# Bone Mineral Density and the Risk of Type-2 Diabetes in Postmenopausal Women: rs4988235 Polymorphism Associated with Lactose Intolerance Effects

**DOI:** 10.3390/nu16173002

**Published:** 2024-09-05

**Authors:** Sylwia Górczyńska-Kosiorz, Edyta Cichocka, Paweł Niemiec, Wanda Trautsolt, Wojciech Pluskiewicz, Janusz Gumprecht

**Affiliations:** 1Department of Internal Medicine, Diabetology and Nephrology, Faculty of Medical Sciences in Zabrze, Medical University of Silesia, 40-055 Katowice, Poland; skosiorz@sum.edu.pl (S.G.-K.); wtrautsolt@sum.edu.pl (W.T.); jgumprecht@sum.edu.pl (J.G.); 2Department of Biochemistry and Medical Genetics, School of Health Sciences in Katowice, Medical University of Silesia, Medykow Street 18, 40-752 Katowice, Poland; pniemiec@sum.edu.pl; 3Metabolic Bone Diseases Unit, Department of Internal Medicine, Diabetology and Nephrology, Faculty of Medical Sciences in Zabrze, Medical University of Silesia, 40-055 Katowice, Poland; wpluskiewicz@sum.edu.pl

**Keywords:** LCT, gene polymorphism, genetic factors, BMI, osteoporosis, diabetes mellitus 2, bone mineral density, lactose intolerance

## Abstract

Dairy products, a major source of calcium, demonstrate a number of beneficial effects, not only protecting against the development of osteoporosis (OP) but also suppressing the onset of type-2 diabetes (T2DM) and improving bone mineral density (BMD). Dairy consumption is closely linked to lactose tolerance. One of the genetic factors predisposing individuals to lactose intolerance is rs4988235 polymorphism of the MCM6 gene. The aim of this reported study was to analyse the relationship between the rs4988235 variant of the MCM6 gene and bone mineral density and the risk of type-2 diabetes in women after menopause. Methods: The study was conducted among 607 female patients in the postmenopausal period in whom bone densitometry and vitamin-D3 levels were assayed and genotyping of the rs4988235 polymorphism of MCM6 gene was performed. The obtained results were analysed for the presence of T2DM, obesity surrogates, medical data, and past medical history. Results: The distribution of genotype frequencies was consistent with the Hardy–Weinberg equilibrium (*p* > 0.050). Postmenopausal women with the GG homozygote of rs4988235 polymorphism consumed significantly less calcium (dairy), which was probably related to the observed lactose intolerance. The GG homozygote of women with rs4988235 polymorphism was significantly more likely to have T2DM relative to the A allele carriers (*p* = 0.023). GG homozygotes had significantly lower femoral–vertebral mineral density despite the significantly more frequent supplementation with calcium preparations (*p* = 0.010), vitamin D (*p* = 0.01), and anti-osteoporotic drugs (*p* = 0.040). The obtained results indicate a stronger loss of femoral-neck mineral density with age in the GG homozygotes relative to the A allele carriers (*p* = 0.038). Conclusions: In the population of women after menopause, the carriage of the G allele of rs4988235 polymorphism of the MCM6 gene, i.e., among the patients with lactose intolerance, significantly increased the risk of developing T2DM and the loss of BMD.

## 1. Introduction

It is recognized that type-2 diabetes is the scourge of the 21st century and it is responsible both for a significant deterioration in the quality of life and reduced survival time due to the development of cardiovascular complications (CVD), heart failure, and chronic kidney disease [1]. In addition to the development of cardiovascular complications, type-2 diabetes with coexisting obesity promotes and exacerbates the development of osteoporosis (OP), particularly among postmenopausal women [2]. In turn, the factors that influence the risk of OP include calcium intake, which affects bone metabolism. Milk and dairy products, which are the main source of calcium, may have a protective effect not only on the development of OP but also on the onset of T2DM. There are some data that dairy consumption can reduce the risk of developing T2DM [3], cardiovascular disease [4], OP [5], and metabolic syndrome (Mets) [6,7]. Consumption of milk and milk products is beneficial for bone mineral density of the lumbar spine and femoral neck [8,9].

Dairy consumption is closely linked to lactose tolerance, i.e., the ability to digest lactose (milk sugar). In people who are lactose intolerant, milk consumption causes gastrointestinal symptoms in the form of bloating and diarrhoea. Lactase is an enzyme that breaks down lactose into galactose and glucose. Lactase, also known as lactase-phlorizin hydrolase, is encoded in humans by the lactase gene *LCT* [10], localised on chromosome 2 (2q21.3). However, lactase expression occurs mainly in the small intestine. During infancy, lactase activity is high but declines with age, resulting in hypolactasia manifested as LNP (lactase non-persistent). LNP occurs in approximately 68% of the global adult population. It is inherited as an autosomal recessive trait [11]. Patients with LNP consume fewer dairy products and demonstrate reduced calcium absorption [11,12,13], thus increasing their risk of developing OP. The retained ability to digest lactose in adults is associated with genetic factors, and patients are referred to as ‘lactase persistent’ (LP) [14,15]. The prevalence of individuals with preserved lactose digestion (LP) varies between populations and is close to zero among Asians and high in populations of European descent.

Localised on chromosome 2q21.3 (a functional single nucleotide polymorphism (SNP)), the rs4988235 variant is located in intron 13 of the *MCM6* (minichromosome maintenance complex component 6) gene and occurs approximately 14 kb upstream of the *LCT* gene. It affects the AP-2 transcription factor binding site [16] and enables an increased transcriptional activation of the LCT lactase gene promoter. The characteristics of the rs4988235 variant are included in Table 1. The presence of the A allele confers the trait of lactose tolerance. The dominant A allele in the homozygous system manifests phenotypically with the ability to digest lactose (LP), and the G allele in the homozygous system with lactose intolerance (LNP), hypolactasia [10]. Heterozygous individuals (referred to as AG) have an intermediate phenotype but are also considered LP [10]. In contrast, in African populations, several additional SNPs confer lactose tolerance (LP) status [15,17]. According to a 2017 meta-analysis covering data from 81% of the global population, the prevalence of lactose intolerance worldwide is 67% (CI 61–72). This prevalence varies regionally, with rates of 64% in Asia (excluding the Middle East); 70% in the Middle East; 47% in Eastern Europe, Russia, and the former Soviet republics; 38% in Latin America; 66% in North Africa; 42% in North America; 63% in Sub-Saharan Africa; and 28% in Europe (excluding Eastern Europe) [18].


A 2021 meta-analysis involving nearly 2 million participants showed that the A allele of the rs4988235 variant associated with higher milk consumption was significantly associated with CVD risk factors such as higher body mass index (BMI) but lower total cholesterol (TC) and LDL-C and HDL-C concentrations [19]. These patients also had a lower risk of developing ischemic heart disease.

**Table 1 nutrients-16-03002-t001:** Genetic characteristics of the variant responsible for the occurrence of lactose intolerance in adults [20,21,22,23].

Characteristics	
gene name	*MCM6* (Minichromosome maintenance complex component 6)
ID SNP/Cytogenetic:	rs4988235, 2q21.3
HGVS	NC_000002.12:g.135851076G>A, Ref. sequence: GRCh38.p14 chr 2
alternative name	NM_002299.2 (LCT): c.-13907C>T, IVS13, C/T
locus and function	The region, located in intron 13 of *MCM6* and surrounding the rs4988235 variant, acts as a promoter for the *LCT* lactase gene. The intron variant is located 13.9 kb upstream of the lactase gene *LCT*
type of inheritance	Autosomal dominant (AD)
allele frequencies	Africans: G 88.2%, A 11.8%; Asians: G 99.3%, A 7%; Caucasians: G 45.7%, A 54.3%
clinical significance	Allele G associated with lactose intolerance, Reported in ClinVar RCV000008124.14
GWAS (genome-wide association study) effects	Allele G: - increases: *Bifidobacterium bifidum* abundance in stool (*p* = 9 × 10^−17^), gut microbiota abundance (phylum Actinobacteria id.400 (*p* = 10^−13^); - decreases: Blood protein levels (*p* = 5 × 10^−210^) and body mass index (*p* = 5 × 10^−9^) Allele A: - increases: lactase-phlorizin hydrolase levels (*p* = 3 × 10^−1451^), body mass index (*p* = 8 × 10^−13^), and hip circumference (*p* = 2 × 10^−8^) - decreases: gut microbiota—bacterial taxa, rank normal transformation method (*p* = 10^−6^)

In view of the incomplete data on the impact of the polymorphism associated with lactose intolerance on the risk of developing type-2 diabetes and bone mineral density, the aim of this reported study was to perform genetic profiling and assess the significance of the rs4988235 variant of the *MCM6* gene associated with the occurrence of lactose intolerance in a group described as RAC–OCT–POL, i.e., randomly selected menopausal women belonging to the population of the Upper Silesia in Poland. We were particularly interested in the effect of the rs4988235 variant on the clinical phenotype of the patients studied, including type-2 diabetes and osteoporosis, BMI, BMD, and other clinical factors.

## 2. Materials and Methods

### 2.1. Study Group

Our study was part of a project called the RAC–OST–92 POL study, which was a retrospective epidemiological cohort study. Our study was approved by the Bioethics Committee of the Medical University of Silesia (No. KNW/0022/KB1/9/I/10) and conducted in accordance with the STROBE guidelines [24].

All the enrolled women provided written informed consent to participate in the study. Six hundred and seven (607) women after menopause with bone mineral density tests were included in the study. In all of them, the allele and genotype frequencies of the rs4988235 polymorphism of the *MCM6* gene were assessed. In the study group, anthropometric data were also collected, vitamin-D3 levels were assayed, and the intake of calcium and vitamin-D3 preparations and the frequency of taking anti-osteoporotic drugs were evaluated. Among the study population, 94 subjects presented with type-2 diabetes. In the subgroup with diabetes, data on diabetes and comorbidities were collected, the history of fractures was obtained, and data on smoking and alcohol consumption were recorded.

### 2.2. Bone Mineral Density Measurement

A Lunar DPX device (GE, Madison, WI, USA) was used to assess bone mineral density. Bone mineral density (BMD) of the non-dominant femoral neck (FN) and hip joint (HJ) was assessed in female patients. BMD was presented in standardised units [g/cm^2^] based on a T-score calculated according to the National Health and Nutrition Examination Survey (NHANES) reference data for white women (aged 20 to 29 years). The WHO criteria were used to diagnose osteoporosis [25]. All the measurements were carried out by a single experienced operator. The coefficient of variation (CV%), calculated from 50 measurements, was 1.6% and 0.82% for FN and HJ, respectively.

### 2.3. Biochemical and Genetic Analyses

A venous blood sample was collected from each participant. After centrifugation of the serum, a two-day column extraction was performed, followed by determination of vitamin-D3 concentration with a 25–OH–Vitamin D ELISA kit (Immundiagnostik, Bensheim, Germany). DNA isolation was performed from blood frozen at −20 degrees C collected on EDTA. A MasterPure DNA kit (Epicenter Technologies, Madison, WI, USA) was used to isolate genomic DNA. Genotyping was based on single nucleotide polymorphism (SNP) determination of the MCM6 gene, C___2104745_20, using the TaqMan Predesigned SNP Genotyping Assay Kits, using a 7300 Real-Time PCR System amplificator (Thermo Fisher Scientific, Vacaville, CA, USA). Genotyping accuracy was checked by re-genotyping 10–15% of the samples. The repeatability of results achieved the 100% level.

### 2.4. Statistical Analysis

Statistical analysis was performed using Statistica 13.1 (TIBCO Software Inc., Santa Clara, CA, USA). The distribution of data was evaluated using the Shapiro–Wilk test. The quantitative data that did not have a normal distribution were presented as medians with quartile deviation (QD). The Mann–Whitney U test was used to represent dichotomous grouping variables. The Krushal–Wallis test, along with post-hoc analysis, was used to compare three or more groups for a given quantitative variable. The relationship between the study variables was assessed using the Spearman’s rank correlation coefficient. The Hardy–Weinberg equilibrium was assessed using the X^2^ test and comparisons of genotype and allele frequencies between groups, differentiated by qualitative variables. For the subgroups with fewer than ten patients, Fisher’s correction was used.

The medians of quantitative variables, such as DXA 141 test score, vitamin-D3 concentrations, and obesity surrogates, were compared in an additive inheritance model (between individual genotypes) and the recessive/dominant model (between the carriers of individual alleles). The qualitative variables (the history of fractures, the diagnosis of osteoporosis according to WHO (T-score ≤ −2.5), the use of anti-osteoporotic therapy, the diagnosis of obesity, diabetes, and others, were assessed for significant statistical differences in genotype frequencies among the classes of these variables.

A *p*-value of less than 0.050 was considered statistically significant. For multiple comparisons, *p*-values were corrected using the Bonferroni correction.

## 3. Results

The general characteristics of the study group are shown in Table 2.

All the women included in the study were in the postmenopausal period. The median number of years after menopause (± QD) was 16.46 ± 6.61. According to the WHO and ISCD criteria, osteoporosis and osteopenia were demonstrated in 9.39% and 55.85% (based on BMD FN T-score) and in 3.79% and 30.15% (based on BMD TH T-score), respectively. Ninety-four (94 women) (15.49%) had type-2 diabetes. The BMI in the study group was 30.83 ± 3.95; BMI ≥ 25 was found in 529 (87.15%) women, and BMI ≥ 30 in 330 (54.37%).

The allele and genotype frequencies of the rs4988235 polymorphism of the *MCM6* gene are shown in Table 3. The distribution of genotype frequencies was consistent with the Hardy–Weinberg equilibrium (*p* > 0.050).

Bone mineral density (BMD FN, BMD TH, BMD TR, along with their T-score values) was analysed in the context of the genotypic variants of the rs4988235 polymorphism of the *MCM6* gene in both the additive and dominant/recessive models. Results from the additive model indicated that GG homozygotes had lower BMD values in each of the parameters analysed relative to the patients with the other polymorphism genotypes (see Figure 1). However, the observed differences did not show statistical significance (*p* > 0.050).

Table 4 summarises the data from the dominant/recessive model. In this model, GG homozygotes had lower bone mineral density than A allele carriers across all the parameters analysed. However, significant differences were only observed for the BMD TR parameter (*p* = 0.037).

There was a negative correlation between the age of the subjects and bone mineral density parameters, separately for GG homozygotes and A allele carriers (*p* < 0.025 in each case). A stronger negative correlation between age and each of the BMD parameters occurred in GG homozygotes relative to A allele carriers (see Table 5), with statistically significant differences only for the BMD FN parameter. The results suggest a stronger loss of femoral-neck mineral density with age in GG homozygotes relative to A allele carriers (*p* = 0.038).

It was checked which qualitative factors had differentiated GG homozygotes and A allele carriers (see Table 6). Among the qualitative factors, GG homozygotes showed a significantly higher prevalence of type-2 diabetes relative to the A allele carriers (*p* = 0.023), significantly more frequent calcium supplementation (*p* = 0.010), vitamin-D3 supplementation (*p* = 0.024) and more frequent anti-osteoporotic treatment (*p* = 0.040) (see Table 6). No similar relationships were shown in the additive model. Vitamin-D concentration did not differentiate between genotype variants, either in the additive or dominant/recessive model (*p* > 0.050). There were no statistically significant differences in vitamin-D concentration between patients with and without diabetes (*p* > 0.050).

## 4. Discussion

Lactose intolerance is a phenomenon that may affect up to two-thirds of the global population. The symptoms of intolerance occur with varying severity and are dependent on a number of factors, such as a reduced expression of intestinal lactase, the amount of lactose in the diet, and the composition of the intestinal flora, which affects the ability to ferment lactose [11]. Research on lactose intolerance has been ongoing for many years, but despite the continued studies, the molecular mechanisms affecting this phenomenon are not fully understood. It is estimated that LNP affects between 31 and 37% of adults in Poland [26].

In the present study, genetic profiling for lactose intolerance was performed by determining the rs4988235 variant of the *MCM6* gene in 607 female patients after menopause. In addition, the clinical phenotype was assessed and correlated with the obtained genotypes. It was found that 87.15% of the menopausal women in the study had a BMI ≥25 kg/m^2^, and 64.09% had abdominal obesity. On the other hand, reduced bone mineral density, allowing for the diagnosis of osteoporosis, was present in 3.79% to 9.39% of the patients (depending on the area studied—trochanter/femoral neck), while osteopenia was found in 30.15% to 55.85% of the women, respectively.

The results of our study remain consistent, in terms of the frequency of the rs4988235 polymorphism of the *MCM6* gene and lactose intolerance, with the results obtained in a group of 63 young Polish adults. Kowalówka M. et al. showed that 33% of the subjects had two G alleles, which was consistent with our results, where 32.46% of the patients were GG homozygotes and 67.54% carried the A allele (AA/AG) [26]. This means that the prevalence of genetic predisposition to lactose intolerance is similar in the entire Polish population and does not only affect menopausal women.

In our study group, it was further observed that women with the GG homozygote of the rs4988235 polymorphism of the MCM6 gene consumed significantly less calcium (milk/sugar), which was probably related to the observed lactose intolerance. Similar findings were noted in a study by Kowalówka M et al., which confirmed that the presence of the GG homozygote of the rs4988235 variant of the MCM6 gene was associated with a significantly lower intake of milk and dairy products [26]. That was also confirmed by an earlier study of the Polish population conducted on a group of 1500 healthy individuals [27]. Popadowska A. et al. showed that the exclusion of lactose from the diet, combined with impaired vitamin-D metabolism, could also lead to an inhibition of calcium absorption, as subjects with adult-type primary lactose intolerance had significantly lower serum vitamin D and calcium concentrations [27].

Based on the data from the Hispanic Community Health Study/Study of Latinos (HCHS/SOL), the association between rs4988235 genotype and milk consumption on T2DM risk was assessed [28]. In that study, there was a balanced distribution of rs4988235 genotypes in the Hispanic population (~60% GG versus ~40% AA/AG) [28]. As expected, the individuals with lactose intolerance (LNP) (i.e., GG homozygotes of the rs4988235 polymorphism) consumed less milk compared to those with confirmed lactose tolerance (LP) (AG/AA). After adjusting for socioeconomic, demographic, and behavioral factors, higher intake was associated with a significantly lower risk of developing T2DM [28].

In our study, 94 women (15.49%) of the study group had type-2 diabetes. An analysis of these data in combination with the prevalence of the G allele showed that the GG homozygote women with rs4988235 polymorphism of the MCM6 gene were significantly more likely to have type-2 diabetes relative to the A allele carriers (*p* = 0.023). Such findings are also supported by a study by de Luis et al., which showed that the presence of the A allele for the rs48988235 variant was associated with a lower risk of T2DM and better metabolism in obese menopausal women. Milk and calcium intake were higher in those women, as were 25-OH vitamin-D concentrations [29].

Previously described studies also demonstrated an association between dairy (milk) intake and some metabolic parameters, i.e., 2 h post-meal glycaemia, fasting insulin levels, triglyceride levels, and obesity prevalence among lactose-intolerant individuals [30,31].

A meta-analysis of six studies conducted in non-Caucasian populations (two studies of the Japanese population, two studies of the Chinese population, one of Puerto Ricans, and one of Native Americans) confirmed that a higher dairy intake could be associated with a reduced risk of developing T2DM (RR = 0.80 (95% CI: 0.66; 0.96) [18,28,32]. Another meta-analysis of 18 studies involving mainly Caucasian patients, i.e., those with a high prevalence of lactose tolerance, found a moderate positive correlation between milk consumption and the risk of T2DM (RR = 1.03 (95% CI: 1.01; 1.04) [11].

When looking for an explanation of the correlation between dairy (milk) intake and T2DM in lactose-intolerant patients, the potential role of the intestinal microflora should be highlighted as a protective factor in this group of patients. Studies have shown that the gut microbiota causes adaptation to lactose intake in people with LNP [19,26]. Despite the beneficial effect of the microbiota, however, it has been found that people with genetically confirmed lactose intolerance consume less lactose (i.e., dairy) on average than lactose-tolerant adults [33]. Probiotics that affect the composition of the intestinal flora may enhance lactose tolerance [34]. A continuous intake of small amounts of lactose has also been shown to lead to an adaptation of the microbiome in persons with LNP and is associated with altered metabolism [33,34]. The beneficial effect of milk intake on the risk of developing T2DM in lactose-intolerant individuals may be partly dependent on several metabolites derived from GAA (γ-glutamylvaline) and tryptophan (indolopropionate) BCAAs (α-hydroxyisocaproate and β-hydroxyisovalerate) [34].

In epidemiological studies assessing the association between milk consumption and the risk of T2DM, the results are not consistent, probably due to the considerable heterogeneity in the populations described, which can partly be attributed to the differences in the prevalence of genotypes of the rs4988235 variant of the MCM6 gene. In their study, Yang et al. did not confirm the association of genetic lactose intolerance with ischemic heart disease, type-2 diabetes, and osteoporosis, but they showed an association with higher fasting insulin levels and BMI [35]. In contrast, Kai Luo et al. demonstrated a protective effect of milk consumption on the risk of developing type-2 diabetes in lactose-intolerant individuals, pointing to a potential involvement of the gut microbiota [29].

However, it should be noted that other factors, such as differences in milk composition, socioeconomic status or other dietary components, gut microbiota, and lifestyle, may also contribute to differences in the association between dairy (milk) intake and the risk of developing T2DM [36,37].

We demonstrate in the presented manuscript that GG homozygotes of the rs4988235 polymorphism of the MCM6 gene in our study had significantly lower femoral vertebral mineral density with respect to the carriers of the A allele (AA and AG genotypes), despite the significantly more frequent supplementation with calcium preparations (*p* = 0.010), vitamin D (*p* = 0.01), and anti-osteoporotic drugs (*p* = 0.040). The results of hip-density measurements were consistent, with the results close to statistical significance (*p* = 0.053). Although no statistically significant differences were found for the femoral-neck measurements, the results indicated a stronger loss of femoral-neck mineral density with age in the GG homozygotes relative to the A allele carriers (*p* = 0.038).

As described earlier, osteoporosis and metabolic diseases such as obesity and diabetes affect microstructural changes in the bone marrow with age and contribute to impaired bone homeostasis, which increases the risk of bone fractures. Therefore, T2DM appears to be an important risk factor for skeletal fragility and osteoporosis in women in post-climacteric period. In the present study, BDM was considered according to the MCM6 gene variant studied. Among GG homozygotes of the rs4988235 polymorphism of the MCM6 gene, i.e., among the patients with lactose intolerance, lower BDMs were described, especially in the vertebral range (BDM TH).

It can be concluded that the presence of the G allele of the rs4988235 polymorphism of the MCM6 gene may be a significant factor in the loss of BDM in women after climacterium and may increase with age. It was also found that the GG homozygotes of the rs4988235 polymorphism of the MCM6 gene were significantly more likely to develop type-2 diabetes; thus, lactose intolerance associated with the MCM6 gene variant studied may be considered a risk factor for the development of type-2 diabetes. Furthermore, GG homozygotes showed significantly more frequent calcium supplementation (*p* = 0.010), vitamin-D3 supplementation (*p* = 0.024), and more frequent anti-osteoporotic treatment (*p* = 0.040), indicating the need to personalize treatment in patients who are GG homozygotes for the variant under study.

One of the strengths of our study presented in this report is the genetic profiling for lactose intolerance in the Polish population of women after climacterium in the context of the clinical profile of the women studied, based on the availability of anthropometric parameters, laboratory results, bone density measurements, physical activity, and comorbidities. The Polish population that we studied was genetically homogeneous [38]. All the women belonged to the Caucasian race and resided in the Upper Silesia region. The study participants were randomly selected and invited to the study from a general population of more than 17,500 women in the post-climacteric period and over 55 years of age. In addition, the group was homogeneous in terms of age (median 65.60 ± 6.28).

The findings obtained in the reported study may allow for the selection of menopausal women at particular risk of developing type-2 diabetes and lower bone mineral density. Our study may also help to individualize treatment by including calcium and vitamin-D supplementation and recommending appropriate diet therapy.

In our study, we did not have access to data on dairy consumption or whether participants consumed lactose-containing products or took lactase with dairy. Therefore, we cannot analyse these data in terms of milk avoidance and the impact of lactose intolerance on dietary choices and bone density.

The study we present in this report lacked information on probiotic intake among the population of female patients after menopause, particularly those with type-2 diabetes, which is undoubtedly a limitation of the study. Nevertheless, it points the way for further research and exploration into the impact of dairy intake on the risk of developing type-2 diabetes.

Neither was the microbiome of both the lactose-tolerant and lactose-intolerant patients studied to determine its impact on the results obtained. In further studies, the use of controlled probiotic supplementation for better control of lactose tolerance would also appear to be advisable.

No follow-up analysis was carried out to better assess the effect of the rs4988235 genotype of the MCM6 gene as a risk factor on the development of complications in type-2 diabetes or the incidence of fractures in the women studied, especially those with reduced bone mineral density [5].

## 5. Conclusions

It was demonstrated that in the population of women after menopause, the carriage of the G allele of the rs4988235 polymorphism of the *MCM6* gene, i.e., among the patients with lactose intolerance, significantly increased the risk of developing type-2 diabetes.

The GG homozygotes of the rs4988235 polymorphism of the *MCM6* gene were also found to have significantly lower femoral vertebral mineral density in relation to the carriers of the A allele (AA and AG genotypes), despite the significantly more frequent supplementation with calcium preparations, vitamin D, and anti-osteoporotic drugs.

There was also a stronger loss of femoral-neck mineral density with age in the GG homozygotes relative to the carriers of the A allele.

The study presented here demonstrates a need for further research to establish a causal relationship between dairy intake, particularly among patients with lactose intolerance, and its impact on the risk of developing metabolic conditions, including type-2 diabetes, in the Polish population. This knowledge will allow for individualization of therapy in groups of patients with type-2 diabetes, osteoporosis, or a high risk of metabolic diseases.

## Figures and Tables

**Figure 1 nutrients-16-03002-f001:**
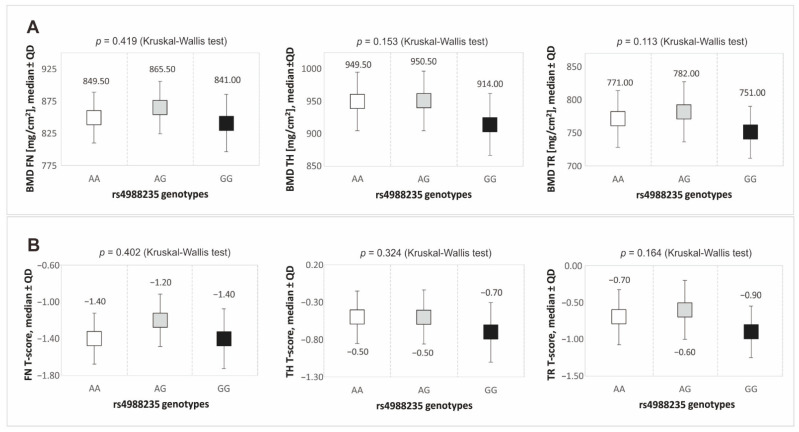
Median values of bone mineral density (**A**) and T-score values (**B**) for the rs4988235 *MCM6* gene polymorphism genotypes. Legend: BMD, bone mineral density; FN, femoral neck; TH, total hip; TR, trochanter; QD, Quartile Deviation.

**Table 2 nutrients-16-03002-t002:** Demographic, clinical, and biochemical characteristics of the study group.

Characteristics			
General	the number of subjects, n (%)	607	(100.00)
	the age [years], median ± QD	65.60	6.28
	the age range [years], n (%)		
	- 50–59	165	(27.18)
	- 60–69	238	(39.21)
	- 70–79	175	(28.83)
	- 80 and more	29	(4.78)
	years after menopause [years], median ± QD	16.46	6.61
	BMI [kg/m^2^], median ± QD	30.82	3.95
	overweight/obesity [BMI ≥ 25], n (%)	529	(87.15)
	obesity [BMI ≥ 30], n (%)	330	(54.37)
	waist circumference [cm], median ± QD	94.00	8.50
	abdominal obesity [WC ≥ 88 cm], n (%)	389	(64.09)
	cigarette smokers, n (%)	71	(11.70)
	alcohol consumption [≥3 units/day], n (%)	4	(0.66)
	calcium supplementation, n (%)	88	(14.50)
	vitamin-D3 supplementation, n (%)	78	(12.85)
	vitamin D3 [ng/mL], median ± QD	7.49	2.76
BMD parameters	BMD FN [mg/cm^2^], median ± QD	855.00	84.5
	BMD FN T-score, median ± QD	−1.30	0.60
	normal BMD [T-score > −1], n (%)	211	(34.76)
	osteopenia [T-score ≤ −1, >−2.5], n (%)	339	(55.85)
	osteoporosis [T-score ≤ −2.5], n (%)	57	(9.39)
	BMD TH [mg/cm^2^], median ± QD	941.00	90.50
	BMD TH T-score, median ± QD	−0.50	0.70
	normal BMD [T-score > −1], n (%)	401	(66.06)
	osteopenia [T-score ≤ −1, >−2.5], n (%)	183	(30.15)
	osteoporosis [T-score ≤ −2.5], n (%)	23	(3.79)
	BMD TR [mg/cm^2^], median ± QD	766.00	87.00
	BMD TR T-score, median ± QD	−0.70	0.75
Comorbidities and medications affecting bone mineral density	DM type 2, n (%)	94	(15.49)
	glucocoticosteroid therapy, n (%)	28	(4.61)
	rheumatoid arthritis, n (%)	40	(6.59)
	thyroid gland diseases, n (%)	6	(0.99)
	chronic kidney disease, n (%)	6	(0.99)

Legend: BMD, bone mineral density; BMI, body mass index; DM, diabetes mellitus; FN, femoral neck; TH total hip; TR, trochanter; QD, Quartile Deviation; WC, waist circumference.

**Table 3 nutrients-16-03002-t003:** Genotype and allele frequencies of the rs4988235 *MCM6* gene polymorphism.

SNP	Position	Genotypes	n (%)	Alleles	n (%)	HWE *p* Value
rs4988235	chr2:135851076	AA	106 (17.46)	A	516 (42.50)	0.370
		AG	304 (50.08)	G	698 (57.50)	
		GG	197 (32.46)			
		AA+AG	410 (67.54)			
		AG+GG	501 (82.54)			

Legend: HWE, Hardy–Weinberg equilibrium; SNP, single nucleotide polymorphism.

**Table 4 nutrients-16-03002-t004:** Bone mineral density (BMD) values for rs4988235 *MCM6* gene polymorphism variants (recessive/dominant model).

Parameter	Median	±QD	Median	±QD	*p* Mann-Whitney U Test
	GG		AA/AG		
BMD FN [mg/cm^2^], median ± QD	841.00	89.50	859.50	80.00	0.297
BMD FN T-score, median ± QD	−1.40	0.65	−1.30	0.60	0.324
BMD TH [mg/cm^2^], median ± QD	914.00	95.50	950.50	91.00	0.053
BMD TH T-score, median ± QD	−0.70	0.80	−0.50	0.75	0.133
BMD TR [mg/cm^2^], median ± QD	751.00	79.00	781.00	89.50	0.037
BMD TR T-score, median ± QD	−0.90	0.75	−0.60	0.80	0.057

Legend: BMD, bone mineral density; FN, femoral neck; TH, total hip; TR, trochanter; QD, Quartile Deviation.

**Table 5 nutrients-16-03002-t005:** Spearman’s correlation coefficient (rs) values for bone mineral density parameters and age of patients in respective genotype variants of the rs4988235 *MCM6* gene polymorphism.

Parameter	GG	AA/AG	*p*
BMD FN [mg/cm^2^], median ± QD	−0.46	−0.33	0.038
BMD FN T-score, median ± QD	−0.44	−0.32	0.054
BMD TH [mg/cm^2^], median ± QD	−0.41	−0.29	0.058
BMD TH T-score, median ± QD	−0.39	−0.30	0.120
BMD TR [mg/cm^2^], median ± QD	−0.30	−0.22	0.162
BMD TR T-score, median ± QD	−0.32	−0.22	0.108

Legend: BMD, bone mineral density; FN, femoral neck; TH, total hip; TR, trochanter; QD, Quartile Deviation.

**Table 6 nutrients-16-03002-t006:** General and clinical characteristics in the variants of rs4988235 *MCM6* gene polymorphism (recessive/dominant model).

Characteristics		GG		AA/AG		
		Median	±QD	Median	±QD	*p* Mann-Whitney U Test
age [years], median ± QD		66.25	5.97	65.47	6.49	0.592
BMI [kg/m^2^], median ± QD		30.13	3.54	31.14	4.07	0.099
waist circumference [cm], median ± QD		92.00	8.00	95.00	9.00	0.194
		n	%	n	%	*p* χ^2^ test
obesity [BMI ≥ 30]	Yes	101	51.27	229	55.85	0.288
	No	96	48.73	181	44.15	
abdominal obesity [WC ≥ 88 cm]	Yes	123	67.96	226	69.82	0.655
	No	58	32.04	115	30.18	
DM type 2	Yes	40	20.30	54	13.17	0.023
	No	157	79.70	356	86.83	
RA	Yes	15	7.61	25	6.10	0.481
	No	182	92.39	385	93.90	
Calcium supplementation	Yes	39	19.80	49	11.95	0.010
	No	158	80.20	361	88.05	
vitamin-D3 supplementation	Yes	34	17.26	44	10.73	0.010
	No	163	82.74	366	89.27	
anti-osteoporotic therapy	Yes	38	19.29	53	12.93	0.040
	No	159	80.71	357	87.07	
history of fractures > 40 year	Yes	54	27.41	115	28.05	0.870
	No	143	72.59	295	71.95	

Legend: BMI, body mass index; DM, diabetes mellitus; RA, rheumatoid arthritis; WC, waist circumference.

## Data Availability

Data are contained within the article.

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
