# Peer review of "Bone Mineral Density and the Risk of Type-2 Diabetes in Postmenopausal Women: rs4988235 Polymorphism Associated with Lactose Intolerance Effects"

_nutrients, 2024, doi:10.3390/nu16173002_

Round 1

Reviewer 1 Report

Comments and Suggestions for Authors

line 33: Dairy is misspelled

lines 43-45 are unnecessary and a scientific distraction

line 62 deficiency is the wrong word.  Please delete.  Your later explanation of lactase non-persistence is correct.

Line 69 Your data cannot distinguish between bone loss and lack of bone development during adolescence.  Hence please eliminate any reference to loss.  Yours is a cross-sectional study, not longitudinal.  

line 84 Apparently there are also middle eastern populations with lactose tolerance.  

line 140 checked out are the wrong words.  evaluated?  

line 226 is this intolerance or LNP?  I think you mean LNP

Do you have dietary data?  there is some evidence that it is milk avoidance and the perception of intolerance that reduces bone density, not the actual genetic make-up.  could you analyze the entire sample by dairy intake and see if you get a larger effect?  ie; are there lactase persistent individuals who think they are intolerant and avoid dairy, thus ending up with lower bone densities.  See Matlik et al Pediatrics 120:3 2017. 

Author Response

I enclose all answers to Reviewer

Reviewer 2 Report

Comments and Suggestions for Authors

Dear Authors,

All research questions are addressed adequately and also correctly responded. Question, of course, deals with the influence of confounding factors, but this doesnt relate to the gene polymorphism.

I have two minor remarks for this manuscript;

1) M+M. please, give for 2.2. WHO criteria the reference and the same for all the methods you have used in the research;

2) glucocorticoid treatment is not a comorbidity (Table 2). Please, re-arrange this under the adequate subtitle.

Otherwise, thanks for the manuscript!

Author Response

I enclosed the answer to Reviewer
